# pADP-ribosylation regulates the cytoplasmic localization, cleavage, and pro-apoptotic function of HuR

Kholoud Ashour[2,3,5,*], Sujitha Sali[1,*], Ali H Aldoukhi[1], Derek Hall[2,3] (ORCID), Souad Mubaid[2,3], Sandrine Busque[2,3], Xian Jin Lian[2,3], Jean-Philippe Gagné[4], Shahryar Khattak[1] (ORCID), Sergio Di Marco[1,2,3], Guy G Poirier[4], Imed-Eddine Gallouzi[1,2,3] (ORCID)

**HuR (ElavL1) is one of the main post-transcriptional regulators that determines cell fate. Although the role of HuR in apoptosis is well established, the post-translational modifications that govern this function remain elusive. In this study, we show that PARP1/2-mediated poly(ADP)-ribosylation (PARylation) is instrumental in the pro-apoptotic function of HuR. During apoptosis, a substantial reduction in HuR PARylation is observed. This results in the cytoplasmic accumulation and the cleavage of HuR, both of which are essential events for apoptosis. These effects are mediated by a pADP-ribose–binding motif within the HuR-HNS region (HuR PAR-binding site). Under normal conditions, the association of the HuR PAR-binding site with pADP-ribose is responsible for the nuclear retention of HuR. Mutations within this motif prevent the binding of HuR to its import factor TRN2, leading to its cytoplasmic accumulation and cleavage. Collectively, our findings underscore the role of PARylation in controlling the pro-apoptotic function of HuR, offering insight into the mechanism by which PARP1/2 enzymes regulate cell fate and adaptation to various assaults.**

## Introduction

The RNA-binding protein (RBP) human antigen R (HuR) is ubiquitously expressed. It contains three highly conserved RNA-binding domains also known as RNA recognition motifs (RRMs; RRM1-3), and a hinge region between RRM2 and RRM3 that serves as the HuR nucleocytoplasmic shuttling (HNS) domain. HuR associates with many mRNA targets by binding mainly to the AU-rich elements (AREs) located in their 3′-UTR, thereby regulating their stability, translation, and/or subcellular localization (Srikantan & Gorospe, 2012; Grammatikakis et al, 2017). HuR is a multifunctional protein that is known to be involved in many cellular processes, including

cell proliferation and differentiation, as well as apoptosis. Our laboratory, as well as others, has previously demonstrated that HuR is required for both pro-survival and pro-apoptotic processes (Lal et al, 2005; Mazroui et al, 2008; Von Roretz et al, 2013). It is known that in response to mild, sublethal stress, HuR modulates the expression of various pro-survival messages such as prothymosin α (Lal et al, 2005). However, when the stress is lethal, HuR shifts its function and modulates the expression of many pro-apoptotic factors such as caspase-9 (Von Roretz et al, 2013). Our group has previously shown that this shift in function is promoted by the caspase-3/7–mediated cleavage of HuR (Von Roretz et al, 2013). Indeed, under apoptotic conditions, HuR translocates to the cytoplasm where it undergoes caspase-dependent cleavage at the aspartate (D)226 residue, thereby generating two cleavage products, HuR-CP1 (24 kD) and HuR-CP2 (8 kD). This event results in the cytoplasmic accumulation of HuR, where HuR mediates its pro-apoptotic function (Von Roretz et al, 2013).

Although HuR is mainly localized to the nucleus under normal conditions, it has the ability to shuttle between the nucleus and the cytoplasm in response to various stimuli. This translocation is mediated via its interaction with protein partners, such as the nuclear export factor PHAPI (also known as pp32) and the nuclear import factor transportin-2 (TRN2) (Mazroui et al, 2008; von Roretz et al, 2011). It has been established that HuR and its cleavage products can be involved in apoptosis via their selective interaction with these protein partners. This interaction defines how HuR modulates the switch of two opposing processes: cell survival and cell death. PHAPI, a well-known HuR ligand, is an important activator of the apoptosome formation. In response to lethal stimuli, the HuR/PHAPI complex is exported from the nucleus to the cytoplasm where both HuR and PHAPI exert their pro-apoptotic function (Mazroui et al, 2008; von Roretz et al, 2011). The cytosolic cleavage of HuR facilitates the release of PHAPI mediating the activation of apoptosome formation. Moreover, we also showed that HuR-CP2 selectively associates with PHAPI, whereas HuR-CP1 interacts with TRN2, resulting in the decreased association of TRN2

[1]KAUST Smart-Health Initiative (KSHI) and Biological and Environmental Science and Engineering (BESE) Division, King Abdullah University of Science and Technology (KAUST), Jeddah, Saudi Arabia  [2]Department of Biochemistry, McGill University, Montreal, Canada  [3]Rosalind & Morris Goodman Cancer Institute, McGill University, Montreal, Canada  [4]Centre de recherche du CHU de Québec-Pavillon CHUL, Faculté de Médecine, Université Laval, Québec, Canada  [5]Faculty of Applied Medical Sciences, Medical Laboratory Technology, Taibah University, Medina, Saudi Arabia

Correspondence: gallouzi.imed@kaust.edu.sa
*K Ashour and S Sali contributed equally to this work

with HuR. The binding of HuR-CP1 to TRN2 therefore blocks the reimport of HuR into the nucleus leading to the accumulation of HuR in the cytoplasm, advancing apoptosis (Mazroui et al, 2008; von Roretz et al, 2011).

The function of HuR, as well as other RBPs, has been shown to be primarily regulated through post-translational modifications, such as phosphorylation, methylation, and more recently poly(ADP-ribosyl)ation (PARylation) (Grammatikakis et al, 2017; Ke et al, 2017). The process of PARylation, whereby polyADP-ribose (PAR) polymers are generated by PAR polymerase enzymes (PARPs), is known, in addition to its fundamental role in DNA repair, to be involved in apoptosis (Pleschke et al, 2000; Wei & Yu, 2016). The PARP family of proteins consists of 17 members that are known to be involved in several cellular processes including PARP1, PARP2, PARP5a (TNKS1), and PARP5b (TNKS2) (Amé et al, 2004; Richard et al, 2021). The most characterized and well-studied enzyme of the PARP family is PARP1. The catalytic activity of PARP1 is normally initiated in response to a break in the DNA strand. When the DNA damage is mild and manageable, PARP1 detects and recruits DNA damage response factors to repair the damage and thereby acts as a cell survival factor. However, when the damage is irreparable, PARP1 is cleaved in the nucleus in a caspase-dependent manner by caspase-3 and caspase-7, thereby leading to apoptosis (Diamantopoulos et al, 2014; Mashimo et al, 2021). Ke et al have shown that the PARylation of HuR, via PARP1, at the D226 residue, regulates its localization and function during inflammation (Ke et al, 2017). They demonstrated that in response to LPS exposure, PARP1 depletion/inhibition decreased the stability of mRNA from pro-inflammatory genes including *Cxcl2* (Ke et al, 2017). Although PARPs can mediate the covalent PARylation of target proteins at specific residues, the catalyzed PAR chain can also bind in a non-covalent manner to proteins that contain a conserved PAR-binding motif (PBM) (Reber & Mangerich, 2021). Generally, this motif consists of a loosely conserved sequence of hydrophobic and basic amino acids, which are often found to overlap with important functional domains such as DNA- and/or RNA-binding domains, exerting regulatory function within the cell (Reber & Mangerich, 2021). Although PARylation of HuR is known to affect its function, the relevance of HuR binding to PAR on its function during apoptosis remains elusive.

Toward the end, in this study, we investigated the role of PARylation in the apoptotic function of HuR. We identified a PBM located in the HNS of HuR. Disruption of this motif results in the cytoplasmic localization of HuR and cell death. We demonstrate, thus, that the interaction of PAR with HuR, through the PBM, plays a key role in mediating the localization and function of HuR during apoptosis.

## Results

### PARP1 and PARP2 regulate the cytoplasmic localization, cleavage, and pro-apoptotic function of HuR

The non-covalent binding of proteins, including RBPs, to PAR, is known to modulate the activity of numerous intracellular pathways including mRNA metabolism and cell death (Krietsch et al, 2013; Teloni & Altmeyer, 2016; Kim et al, 2020). To establish the role of PARylation on the function of HuR during apoptosis we assessed, as a first step, whether HuR is associated with PAR polymers in HeLa cells treated, over 3 h, with staurosporine (STS), a well-known apoptotic inducer. We have previously demonstrated, as described in Mazroui et al (2008) and shown in Fig S1A and B, that the treatment of these cells with 1 µM of STS, over a 3-h period, increases the cytoplasmic accumulation of HuR, resulting in the cleavage of HuR and PARP1, as well as the activation of apoptotic pathways (as evidenced by the cleavage of the effector caspase, caspase-3). We show, under these conditions, by performing co-immunoprecipitation experiments, that although PAR associates with HuR in untreated cells, this interaction decreases when the cells were treated with this lethal dose of STS for up to 3 h (Fig S1C). We next assessed whether the association of PAR with HuR was correlated with the severity of the stress (mild, moderate, and lethal). We did so by treating cells with different concentrations (0.1, 0.25, 0.5, and 1 µM) of STS for 3 h which increases, in a dose-dependent manner, the cleavage of HuR and PARP1, as well as the activation of apoptosis (as assessed by the cleavage of caspase-3) (Fig 1A). These results showed that although treatment with lower doses of STS, such as 0.1 and 0.25 µM, mimicked mild stress conditions higher doses, such as 0.5 and 1 µM, were considered as moderate and lethal conditions, respectively. The cytoplasmic accumulation of HuR, under these conditions, correlated with the cleavage of PARP1 and HuR (Figs 1B and S2). Although HuR remained nuclear in cells treated with 0.1 and 0.25 µM STS, it began to become cytoplasmic upon treatment with 0.5 µM. Interestingly, we also observed that although HuR is associated with PAR in untreated cells and cells treated with 0.1 and 0.25 µM of STS, the association was lost when the cells were treated with higher, more lethal doses of STS (0.5 and 1.0 µM) (Fig 1C).

Epigallocatechin-3-gallate (EGCG) has been previously shown to induce apoptosis in a variety of cancer cells (Chu et al, 2017; Khiewkamrop et al, 2018). One way it does so is by targeting PARP16, resulting in the enhanced activation of ER stress–induced apoptosis (Wang et al, 2017). We thus assessed whether treatment of cells with EGCG affects the association of HuR with PAR and whether this effect was correlated with the cytoplasmic accumulation of HuR, its cleavage and, in addition, the induction of apoptosis. We observed that despite the activation of apoptosis in these cells, HuR remained localized in the nucleus associated with PAR (Fig 1A–C). Collectively, our results indicate that the decreased interaction of HuR with PAR coincides with the cleavage of HuR and PARP1, as well as the activation of apoptosis (Figs 1A and C and S1A and C). PARP1 therefore prevents the induction of apoptosis by promoting the interaction of HuR with PAR.

As mentioned earlier, our laboratory has shown that in response to lethal stress, the accumulation of HuR in the cytoplasm is required for its cleavage and pro-apoptotic function (Fig S1A and B) (Mazroui et al, 2008). Because the interaction of HuR with PAR occurs in untreated cells as well as cells exposed to mild stress conditions (where HuR is localized in the nucleus) (Figs 1A–C and S1A–C), we decided to assess whether PARP1-mediated PARylation prevents the accumulation of HuR in the cytoplasm. PARP2 has a highly redundant function to PARP1, and both are cleaved in a

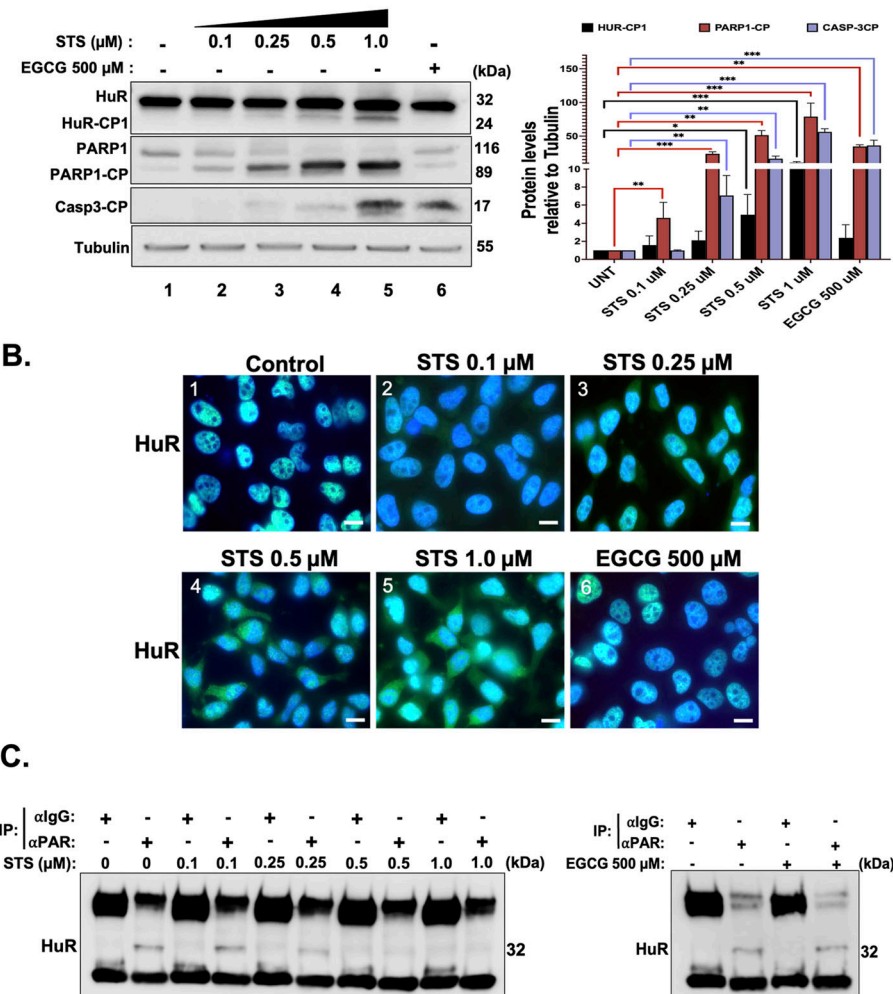

**Figure 1. Cytoplasmic translocation and cleavage of HuR in response to an apoptotic stimulus correlate with the cleavage of PARP1/2.**
**(A)** (Left) HeLa cells treated with or without different concentrations of STS (0.1, 0.25, 0.5, and 1 µM) or 500 µM EGCG for 3 h were collected, lysed, and used for Western blot analysis with antibodies against HuR, PARP1, cleaved caspase-3, or α-tubulin (loading control). (Right) Densitometric quantification of HuR-CP1, PARP1-CP, and cleaved caspase-3 signals in the Western blot relative to the α-tubulin signal. Values were quantified using ImageJ.
**(B)** Immunofluorescence experiments demonstrating the localization of HuR in HeLa cells treated with or without different concentrations of STS (0.1, 0.25, 0.5, and 1 µM) or 500 µM EGCG for 1.5 h. The images shown are a merge of the HuR and DAPI (staining for nuclei) signals. **(C)** Lysates obtained from HeLa cells treated with or without different concentrations of STS (0.1, 0.25, 0.5, and 1 µM) (left) or 500 µM EGCG (right) for 1.5 h were used for immunoprecipitation experiments using antibodies against PAR or IgG as a negative control. The binding of PAR to HuR was then assessed by Western blot using an anti-HuR antibody (3A2). All blots shown in the figure are representative of three independent experiments. Data presented in Fig 1 are ± the S.E.M. of three independent experiments with *P < 0.05, **P < 0.01, and ***P < 0.001 by an unpaired t test.
Source data are available for this figure.

caspase-dependent manner during apoptosis (Benchoua et al, 2002; Ali et al, 2016). Thus, we sought to assess whether depleting both PARPs individually and in combination, using siRNAs specifically targeting each PARP, would affect the localization of HuR. We observed that these siRNAs efficiently depleted the expression of both PARPs by more than 90% in these cells (Fig S3A). By performing immunofluorescence (Fig 2A) and subcellular fractionation coupled to Western blot experiments (Fig S3B), we observed that although knocking down PARP1 increased the cytoplasmic localization of HuR in untreated conditions, this effect was more prominent when cells were simultaneously treated with siRNAs targeting both proteins. This observation was further supported by data showing that treatment of cells with talazoparib, a well-known PARP1/2 inhibitor, increased the cytoplasmic accumulation of HuR (Figs 2B and S3C). Our results therefore suggest that the retention of HuR in the nucleus of untreated HeLa cells is mediated by the PARP1/2-induced interaction of HuR with PAR.

Next, to determine the impact of PARP1 and PARP2 on the apoptotic function of HuR we assessed whether knocking down

these PARPs affects its cleavage. We noticed that HuR cleavage is increased in PARP1-depleted cells under normal conditions (Fig 2C, lane 2). This cleavage, however, was further significantly increased with the double knockdown of PARP1 and PARP2 compared with siCtl-treated conditions (Fig 2C, lane 4). These results thus indicate that the PARylation of HuR could play a potential role in modulating its pro-apoptotic function. Since the depletion of these PARPs resulted in the cleavage of HuR in untreated conditions, mimicking what we observed in the apoptotic conditions, we next questioned the impact of depleting these PARPs on caspase-3 cleavage, another well-established event in apoptosis. As expected, simultaneously silencing PARP1 and PARP2 resulted in a significant increase in the cleavage of caspase-3 (Fig 2C, lane 4). This result was further confirmed by performing flow cytometry experiments which demonstrated a significant increase in the number of annexin V–positive cells under siPARP1- and siPARP2-treated conditions (Fig 2D). Together, these findings highlight the importance of the PARP1/2-mediated PARylation of HuR on its function during apoptosis.

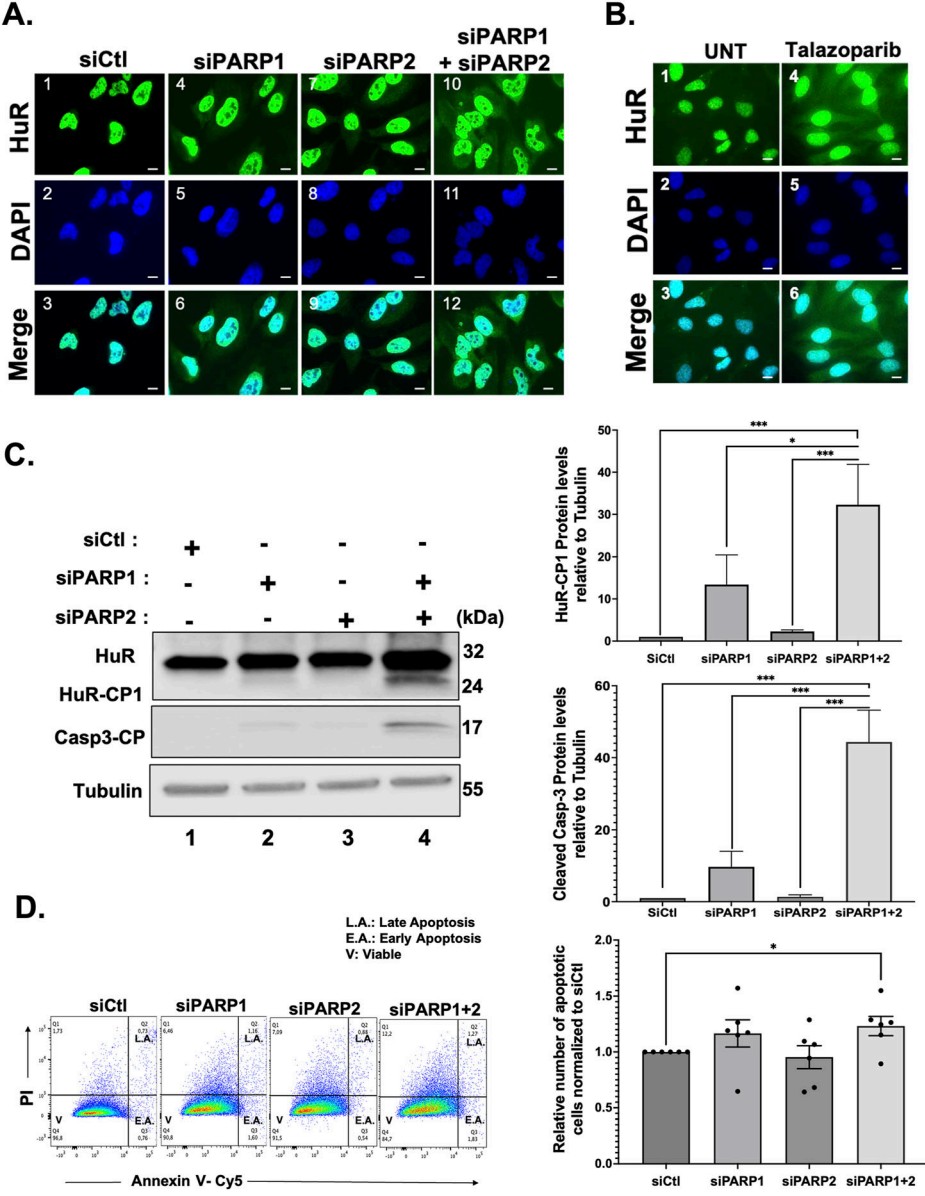

**Figure 2. PARP1/2 knockdown increases HuR cleavage and triggers apoptosis.**
**(A)** HeLa cells were transfected with siRNA targeting PARP1 and/or PARP2 or a control siRNA. These cells were then fixed, permeabilized, and stained with antibodies against HuR. DAPI was used to stain nuclei. Images are representative of three independent experiments. Scale bars, 10 $\mu m$. **(B)** Immunofluorescence experiments demonstrating HeLa cells treated with and without 1 μm talazoparib. After 24 h of treatment cells were fixed, stained, permeabilized, and stained with antibodies against HuR. DAPI was used to stain nuclei. Images of a single representative field are shown and are a representation of three independent experiments. Scale bars, 10 $\mu m$. **(C)** HeLa cells were transfected with siRNA targeting PARP1 and/or PARP2 or a non-specific control siRNA (siCtl). Lysates were used for Western blot analysis (left panel) with antibodies against HuR, caspase-3 cleavage product (CP), and α-tubulin. Densitometric quantification (right panel) of HuR-CP1 and caspase-3-CP levels. Values were quantified using ImageJ, normalized to tubulin, and shown relative to siCtl. **(D)** HeLa cells treated with siRNA as described above were analyzed by staining with annexin V–Cy5 and PI (propidium iodide) and by flow cytometry. The relative number of apoptotic cells was determined for siPARP1 and/or siPARP2. The values are relative to control siRNA-treated cells. Data presented in Fig 2 are ± the S.E.M. of three independent experiments with *$P < 0.05$ and ***$P < 0.001$ by an unpaired $t$ test. Source data are available for this figure.

## PAR binds HuR non-covalently through a consensus motif

It is well established that PARP-mediated PARylation of target proteins can occur either by covalent modifications or by non-covalent association with PAR (Gagne et al, 2008). Both mechanisms were shown to entail different functional consequences on the affected proteins (Ji & Tulin, 2013; Duan et al, 2019). Thus, as a first step, we decided to investigate whether the non-covalent association of PAR with HuR could mediate its pro-apoptotic function. By performing an in vitro dot-blot assay, we demonstrated that HuR, unlike BSA and GST (used as negative controls), non-covalently binds to PAR (Fig 3A). Next, we wanted to determine the exact PAR-binding site on HuR. To this end, we performed an in vitro peptide mapping experiment where we generated small peptide fragments

spanning the complete HuR sequence. Each fragment is about 20 amino acids in length. We found that several fragments (B6, B7, E1) of HuR exhibited binding to PAR with various strengths (Fig 3B). However only one of these (E1) harbors a region (amino acids 201–208 of HuR) that exhibits 76% similarity to a well-known consensus PAR-binding site ($[HKR]_1$-$X_2$-$X_3$-$[AIQVY]_4$-$[KR]_5$-$[KR]_6$-$[AILV]_7$-$[FILPV]_8$) (Pleschke et al, 2000; Gagne et al, 2008). Therefore, we dubbed this element as the HuR PAR-binding site (HuR-PBS).

To better understand the importance of this site on the function of HuR, we generated a mutant isoform of HuR (HuR$^{PBmt}$) whereby the (+)-charged arginine (R) and histidine (H) residues within the HuR-PBS were converted into alanines (A) (Fig 3C, left). Using the dot-blot approach mentioned above, we demonstrated that unlike the wild-type HuR (HuR$^{wt}$), the HuR PAR-binding mutant (HuR$^{PBmt}$)

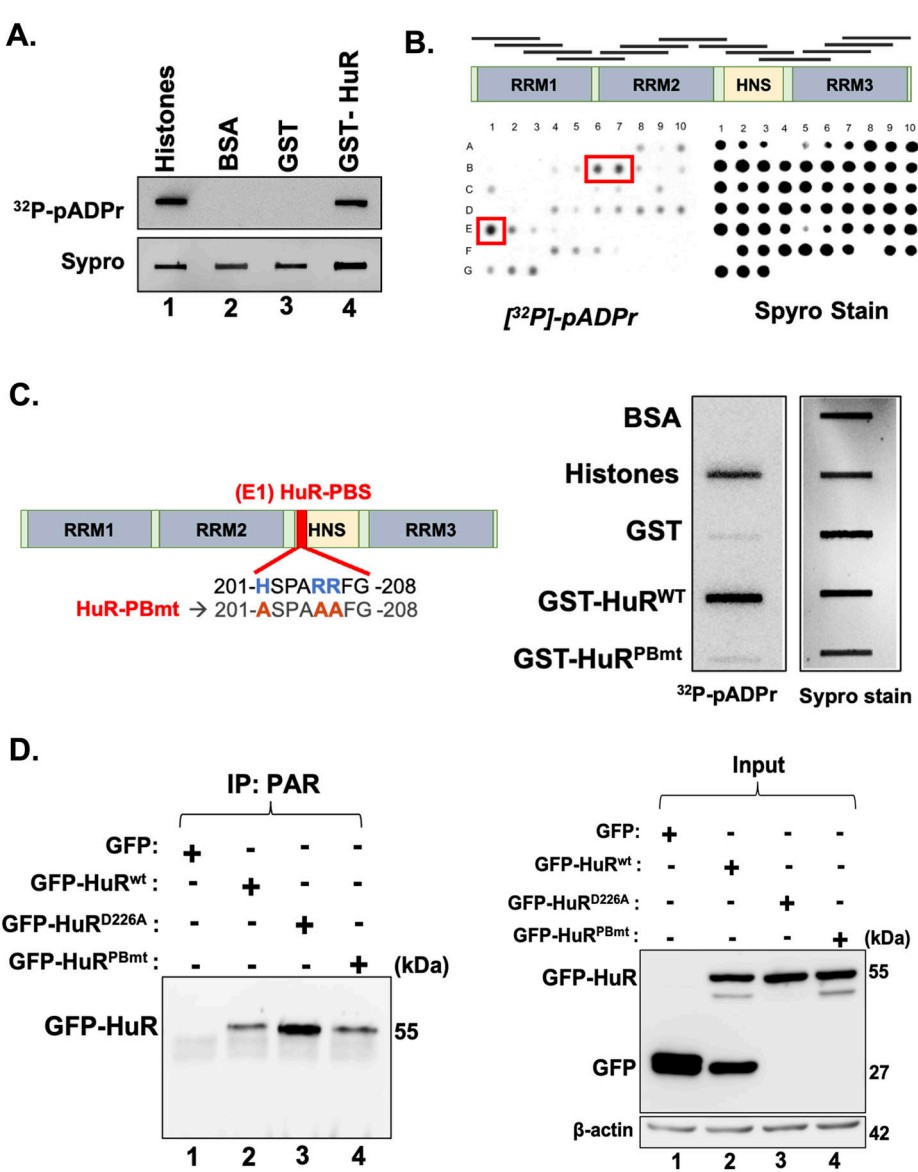

**Figure 3. PAR binds HuR non-covalently through a consensus motif.**
**(A)** Recombinant GST and GST-HuR proteins, as well as recombinant histone (positive control) and BSA (negative control), were blotted directly onto a nitrocellulose membrane, rinsed, incubated with a radiolabeled $^{23}$P-pADPr, and analyzed by autoradiography. The SYPRO Ruby stain was used to demonstrate the integrity and quantity of the proteins. **(B)** HuR protein was fragmented into 63 small peptides (each fragment is 20 amino acids in length with 5 staggered amino acids) used for the peptide mapping experiment. All fragments were blotted onto a nitrocellulose membrane and processed as in (A). **(C)** (Left) Schematic showing the location of the PAR-binding site of HuR. Mutation of this site was generated by substituting the positively charged amino acids histidine and arginine by hydrophobic alanine residues. (Right) Slot-blot assay was performed using recombinant GST-HuR$^{wt}$, GST-HuR$^{PBmt}$ protein, and GST/BSA as a negative control, while histone as a positive control. SYPRO Ruby stain was used to demonstrate the integrity and quantity of the proteins. **(D)** (Left) Total cell extracts obtained from HeLa cells transfected with GFP, GFP-HuR$^{wt}$, GFP-HuR$^{PBmt}$, and GDP-HuRD226A were used for immunoprecipitation experiments using antibodies against PAR. The binding of PAR to HuR was then assessed by Western blot using anti-GFP antibody. (Right) Transfection efficiency was assessed by determining the levels of these proteins in the input using anti-GFP and $\beta$-actin as a loading control. Immunoprecipitation results are representative of three independent experiments. Source data are available for this figure.

lost its ability to bind PAR (Fig 3C, right). The mutation of the PBS did not, however, affect the RNA-binding function of HuR. We show, by performing RNA immunoprecipitation (RNA-IP) experiments, that both HuR$^{wt}$ and HuR$^{PBmt}$ were similarly associated with the previously identified mRNA targets *caspase-9* and *prothymosin α* (ProTα) in HeLa cells (Fig S4).

HuR has also been previously reported to be covalently PARylated at the D226 residue (Ke et al, 2017). We thus assessed whether the mutation of the D226 residue affects the non-covalent PARylation of HuR and, vice versa, whether the mutation of the PBM of HuR affects its covalent PARylation. Toward this end, we immunoprecipitated PAR from HeLa cells transfected with GFP, GFP-HuR$^{wt}$ (with the GFP added to the N-terminal of HuR), GFP-HuR$^{PBmt}$, and GFP-HuR$^{D226A}$ (containing a mutation at the D226 residue), followed by Western blot analysis with anti-GFP (Fig 3D, left panel). We demonstrated, by performing these experiments, that GFP-

HuR$^{wt}$ and GFP-HuR$^{PBmt}$ were similarly associated with PAR, suggesting that the mutation of the PBM did not hinder the covalent PARylation of HuR. Mutation of the D226 residue, on the contrary, increased the non-covalent association of PAR with HuR. This effect is likely due to the fact that the GFP-HuR$^{D226A}$ mutant, while not cleaved, prevents caspase-mediated apoptosis (Fig 3D, right panel). Together, these results reveal that HuR non-covalently interacts with PAR through the harbored PBS in HeLa cells under normal conditions.

## PAR binding prevents the pro-apoptotic function of HuR by promoting its nuclear localization

Our results described above show that the depletion of PARP1/2 decreased the nuclear localization of HuR. This is likely due to the decreased interaction of HuR with PAR. To assess whether this is the

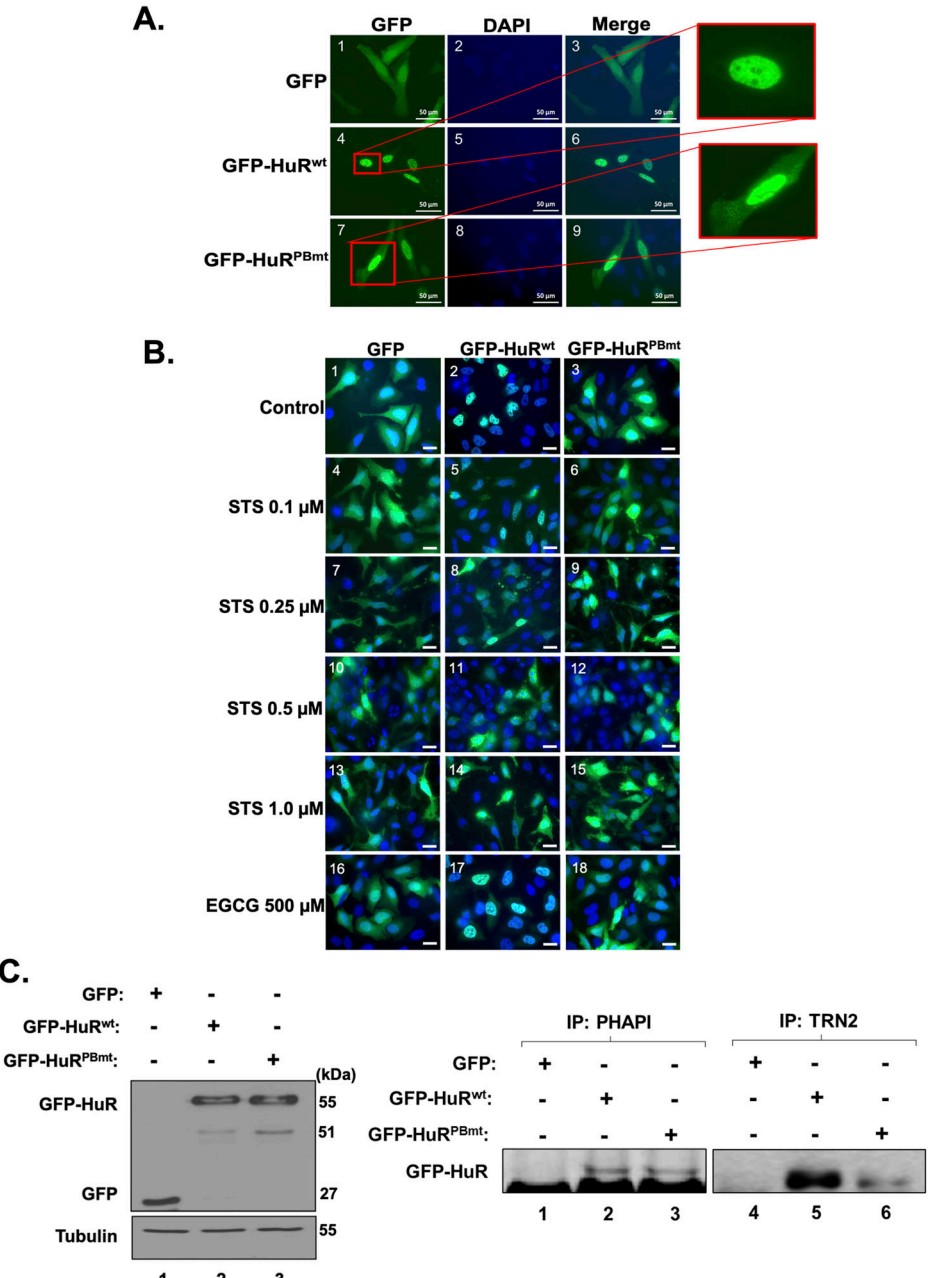

**Figure 4. HuR binding to PAR modulates its cellular localization in HeLa cells.**
**(A)** HeLa cells transfected with GFP, GFP-HuR[wt], and GFP-HuR[PBmt] were fixed, permeabilized, and stained with antibodies against HuR and DAPI. Images are representative of three independent experiments. Scale bars, 50 μm.
**(B)** Immunofluorescence experiments were performed with HeLa cells transfected with GFP, GFP-HuR[wt], and GFP-HuR[PBmt] and treated with or without different concentrations of STS (0.1, 0.25, 0.5, and 1 μM) or 500 μM EGCG for 1.5 h. The images shown are a merge of the GFP and DAPI (staining for nuclei) signals. Scale bars, 10 μm.
**(C)** Total cell extracts obtained from HeLa cells transfected as in (A) were used for immunoprecipitation experiments using antibodies against PP32/PHAPI (left panel) or TRN2 (right panel). The immunoprecipitated complex was then assessed by Western blot using an anti-GFP antibody. The blots are representative of three independent experiments. Source data are available for this figure.

case, we next assessed the impact of mutating the HuR PAR-binding site on its cellular localization. Immunofluorescence assays and subcellular fractionation experiments revealed that GFP-HuR[PBmt] but not GFP-HuR[wt] accumulates in the cytoplasm of untreated HeLa cells, mimicking the observations obtained with the knock-down of PARP1/2 (Figs 4A and S5). Interestingly, although the dose-dependent treatment of cells with STS did not affect the cellular localization of GFP-HuR[PBmt] (which remained cytoplasmic), it did, similar to endogenous HuR (Figs 1B and S2), lead to the increased cytoplasmic accumulation of GFP-HuR[wt] (Figs 4B and S6). GFP-HuR[wt], interestingly, remained localized to the nucleus

of cells treated with EGCG. Together, our results therefore suggest that the non-covalent association of PAR with HuR plays an important role in modulating its cellular localization in normal HeLa cells.

We and others have shown that the nucleocytoplasmic trans-location of HuR, during apoptosis, is mediated by its association with adaptor proteins for nuclear export such as PHAPI and with import factors such as transportin-2 (Brennan et al, 2000; Mazroui et al, 2008; Zhang et al, 2016). To determine whether mutating the PAR-binding site would have an impact on the differential asso-ciation of HuR with these proteins, we immunoprecipitated PHAPI

## A.

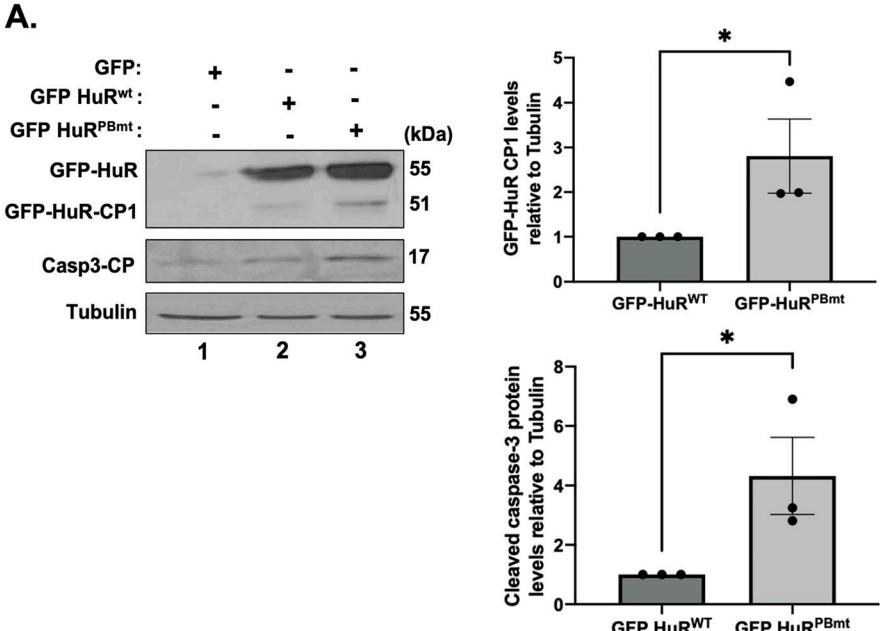

**Figure 5. PAR binding to HuR negatively affects its pro-apoptotic function.**
**(A)** HeLa cells were transfected with GFP, GFP-HuR[wt], and GFP-HuR[PBmt]. (Left) Lysates were used for Western blot analysis with antibodies against GFP, caspase-3-CP, and α-tubulin. (Right) Densitometric quantification of GFP-HuR-CP1 and caspase-3-CP levels was normalized to α-tubulin level and shown relative to GFP-HuR[wt]. **(B)** HeLa cells transfected as described in (A) were analyzed by staining with annexin V–Cy5 and PI and by flow cytometry. The relative number of apoptotic cells was determined for GFP-HuR[wt]– and GFP-HuR[PBmt]–transfected HeLa cells. The values for GFP-HuR[PBmt] were normalized to GFP-HuR[WT] levels. Data presented in Fig 5 are ± the S.E.M. of three independent experiments with *P < 0.05 and ***P < 0.001 by an unpaired t test. Source data are available for this figure.

## B.

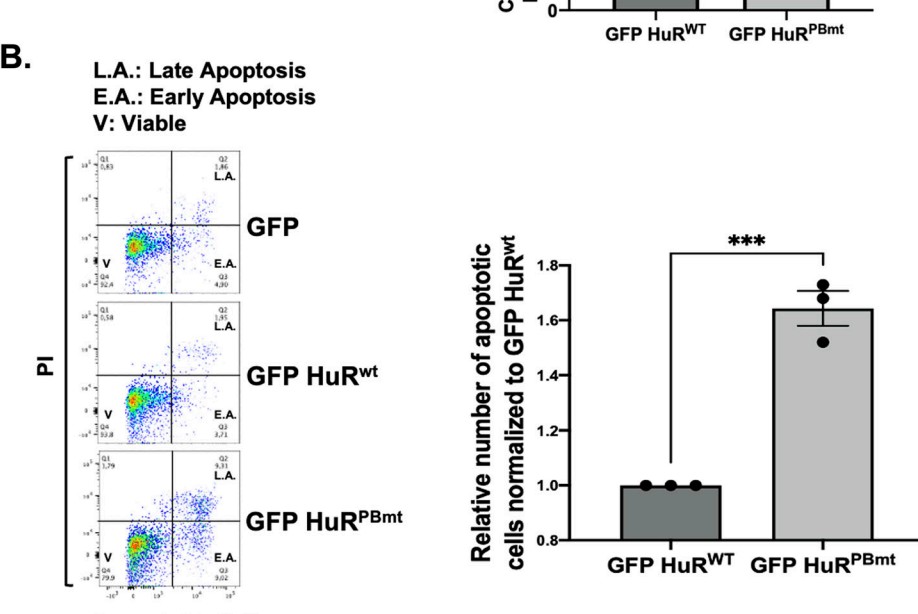

and TRN2 individually and assessed their association with GFP-HuR[wt] or GFP-HuR[PBmt] (Fig 4C). We observed that unlike HuR[wt], the binding of the HuR[PBmt] isoform to TRN2 but not to PHAPI is reduced in untreated cells. This finding suggests that an intact HuR-PBS is required for the association of HuR with TRN2 and its retention in the nucleus. We have previously shown that the cleavage of HuR is tightly related to its cytoplasmic accumulation because of the competition of HuR-CP1 with full-length HuR for the binding to TRN2, leading to the accumulation of full-length HuR in the cytoplasm. Therefore, we next determined whether mutating the HuR-PBS would affect the cleavage of HuR. We observed that the GFP-HuR[PBmt] is cleaved to a greater extent than GFP-HuR[wt] (Fig

5A). Interestingly, we show that the expression of HuR[PBmt] increased the cleavage of caspase-3 to a greater extent than cells expressing HuR[wt] (Fig 5A). To determine the physiological importance of PAR binding, we performed flow cytometry analysis to assess the cell fate of HuR[PBmt]-expressing cells compared with cells expressing HuR[wt]. These results further supported our findings described above and showed an increase in annexin V–positive cells expressing HuR[PBmt] (Fig 5B), providing evidence for the anti-apoptotic role conferred to HuR by binding to PAR. In summary, we demonstrate that HuR can non-covalently bind to PAR and that this binding inhibits its pro-apoptotic function under normal conditions.

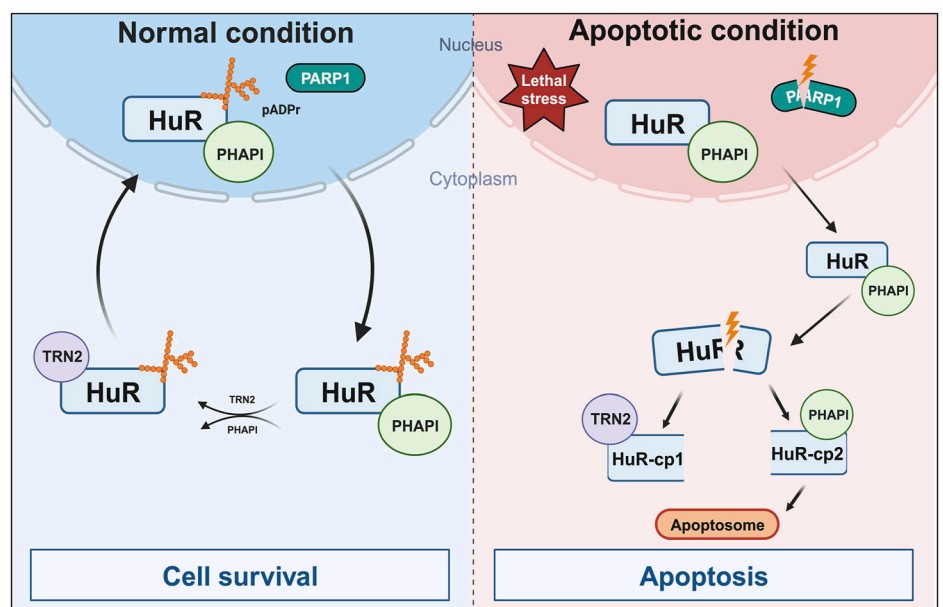

**Figure 6. Proposed model.**
Model depicting the mechanism by which HuR association with PAR polymers regulates its apoptotic function. Under normal conditions, HuR interacts with PAR polymers through its PAR-binding site (HuR-PBS) maintaining its nuclear localization by promoting its interaction with the import factor TRN2. In response to a lethal assault, HuR loses its binding to PAR concurrently with the cleavage of PARP1. HuR/PHAPI translocate to the cytoplasm where HuR undergoes caspase-mediated cleavage yielding HuR-CP1 and HuR-CP2. Although HuR-CP2 interacts with PHAPI mediating the activation of apoptosome formation, HuR-CP1 interacts with TRN2 preventing the reuptake of HuR back to the nucleus. HuR therefore accumulates in the cytoplasm, advancing apoptosis.

## Discussion

In this study, we identify PARylation as a regulatory mechanism that modulates the function of HuR in determining cell fate. Our results show that PARP1/2-mediated PARylation prevents the accumulation of HuR in the cytoplasm resulting in a decrease in its cleavage and inhibition of HuR's pro-apoptotic function. We demonstrated that the combined depletion of PARP1 and PARP2 increases the cytoplasmic accumulation of HuR resulting in its increased cleavage. HuR cleavage, consequently, increases its pro-apoptotic function as evidenced by the significant increase in the level of caspase-3 cleavage and in the number of apoptotic cells. Furthermore, we showed that PAR binds HuR non-covalently through a consensus motif and that this binding is required for the nuclear localization of HuR, as well as its association with the import factor TRN2. Indeed, we found that mutating HuR-PBS prevented PAR from binding to HuR, resulting in its cytoplasmic accumulation and the activation of apoptosis. Thus, our work provides evidence for the importance of the PARP-mediated PARylation and the resulting PAR binding to HuR in regulating the function of HuR during apoptosis (Fig 6).

Although PARylation of HuR has been previously shown to regulate the function of HuR during inflammation (Ke et al, 2017, 2021), as well as muscle cell differentiation (Mubaid et al, NAR, accepted for publication), the importance of this modification on HuR function in cell fate was not assessed. Indeed, recent studies by Ke et al revealed that in response to inflammatory stimuli, PARP1-mediated PARylation of HuR occurs on the aspartate residue 226. They demonstrated that mutating this site (D226) or inhibiting PARP impacted HuR localization, its ability to associate with pro-inflammatory messages, and its oligomerization (Ke et al, 2017, 2021). More recently, our laboratory uncovered that PARylation of HuR by tankyrase1 (TNKS1), also known as PARP5a, promoted HuR

cytoplasmic accumulation and cleavage, as well as its ability to associate with promyogenic mRNAs during myogenesis (Mubaid et al, NAR, accepted for publication). In this study, however, we identified a consensus PBM located within the HNS of HuR, and we showed that PARP1/2-mediated PARylation and PAR binding to HuR through this identified motif mediates its subcellular localization and function during apoptosis.

Although HuR is predominantly a nuclear protein under normal conditions, its HNS domain encompasses a nucleocytoplasmic shuttling sequence allowing it to shuttle between the nucleus and the cytoplasm in response to various stimuli, such as stress signals (Fan & Steitz, 1998; van der Giessen & Gallouzi, 2007; Von Roretz et al, 2013; Grammatikakis et al, 2017). This translocation is important for the HuR-mediated post-transcriptional regulation of many mRNA targets including mRNA localization, stabilization, and translation, and has been shown to have physiological relevance by affecting cell fate and muscle cell differentiation (Brennan & Steitz, 2001; van der Giessen & Gallouzi, 2007; Mazroui et al, 2008; von Roretz et al, 2011). Several studies have reported that post-translational modification of residues within the RRMs influences the function of HuR in regulating RNA metabolism, whereas modification of residues within or near the HNS impacts HuR's subcellular localization (Gallouzi & Steitz, 2001; Srikantan & Gorospe, 2012; Grammatikakis et al, 2017). For example, phosphorylation of HuR by Chk2 at HuR residues S88, S100, and T118 located within RRM1 and RRM2 modulates HuR binding to SIRT1 mRNA and other mRNA targets (Abdelmohsen et al, 2007). On the contrary, phosphorylation by cdk1 at S202 facilitates HuR binding to the nuclear 14-3-3 triggering its nuclear retention (Kim et al, 2008; Grammatikakis et al, 2017). Previously, our laboratory and others have shown that the localization of HuR is dependent on its HNS, which mediates the differential association of HuR with protein partners for nuclear export, such as PHAPI and APRIL, and with the import factors TRN-1-

2 and importin α (Brennan et al, 2000; Rebane et al, 2004; Mazroui et al, 2008). It has been shown that under normal conditions, HuR is localized mainly to the nucleus (Mazroui et al, 2008). However, in response to lethal stress, HuR and PHAPI translocate to the cytoplasm where it is cleaved in a PKR-dependent manner by caspase-3 and caspase-7 yielding two cleavage products (HuR-CP1 and HuR-CP2) (Mazroui et al, 2008; Beauchamp et al, 2010; von Roretz & Gallouzi, 2010; Von Roretz et al, 2013). Moreover, our laboratory showed that HuR-CP1 associates with TRN2 preventing the nuclear reuptake of HuR, thus causing HuR to accumulate in the cytoplasm (von Roretz et al, 2011). In this present study, we identified the non-covalent binding of PAR to HuR as a regulatory mechanism mediating its association with these partners and therefore its pro-apoptotic function. Given that the HuR-PBS is located in the HNS, it is not surprising that it regulates the localization of HuR. Our results demonstrate that an intact PAR binding to HuR is required for its binding with TRN2 in particular and that mutating this site resulted in the loss of this binding. As HuR binding to different protein partners seems to influence the function of HuR during apoptosis, it would be valuable to determine whether mutating the PBS on HuR would have an impact on HuR association with different protein ligands. Indeed, the possibility exists that the cellular localization and/or expression of HuR, during apoptosis, could be independent of TRN2 and involve the non-covalent binding of HuR to these other protein ligands. Also, as previous observations highlighted the importance of HuR cleavage products during the onset of apoptosis, it would be interesting to investigate the role of PARP-mediated modification of HuR and/or binding to PAR on the role of HuR-CPs as pro-apoptotic players.

In addition to the ability of PARPs to covalently modify acceptor proteins at specific residues, a number of proteins, also known as PAR readers, can be modified by the non-covalent binding of PAR to consensus PBMs (Kamaletdinova et al, 2019). The mechanism through which these motifs bind non-covalently to PAR is, however, not well understood. One hypothesis that has been proposed is that proteins containing a PBM interact with PAR by binding between the second phosphate of one ADP-ribosyl moiety and the first phosphate of the next (Kamaletdinova et al, 2019; Reber & Mangerich, 2021). This would suggest that HuR, thus, containing a PBM, most likely binds to ADPr polymers rather than single PAR molecules. This is supported by our in vitro data demonstrating that mutations of this motif prevented the binding of HuR to ADPr polymers generated by automodified PARP1 (Fig 3C).

Many RBPs have been shown to be bound by PAR covalently and non-covalently, both of which lead to the alteration of their functions (Kamaletdinova et al, 2019). Previous studies have reported that HuR is covalently PARylated by PARP1 at the D226 residue in LPS-induced cells, which affected its localization and function (Ke et al, 2017, 2021). Moreover, HuR also has been shown, in a proteome-wide analysis of PAR-associated proteins, to bind PAR non-covalently (Jungmichel et al, 2013). In our study, we confirmed that PAR binds HuR non-covalently and we identified the consensus PAR-binding site and showed its physiological importance in the anti-apoptotic function of HuR. Although the covalent PARylation of HuR at the D226 residue is critical in regulating its function and localization in macrophages (Ke et al, 2017), the role of

this modification on HuR function in normal HeLa cells remains unknown (Fig 3D).

Many recent studies are now pointing to the importance of these two manners of PARylation on the function of their substrates and how there may be an interplay between the two modifications (Grammatikakis et al, 2017; Alemasova & Lavrik, 2019; Ke et al, 2019). For instance, heterogeneous nuclear ribonucleoprotein A1 (hnRNPA1), a well-known RBP, has been shown to be PARylated covalently and it can also bind to PAR or PARylated proteins non-covalently. Recently, Duan et al showed that hnRNPA1 is PARylated on lysine 298 and mutating this site decreased its PARylation and affected its localization (Duan et al, 2019). They also showed that when the PBM is mutated, it increased its covalent PARylation (Duan et al, 2019). These observations led them to suggest that the non-covalent PAR binding reduces the hyper-PARylation of hnRNPA1. Importantly, this study shows that mutating the non-covalent PARylation-binding site prevented the oligomerization of hnRNPA1 and prevented the formation of stress granule (Duan et al, 2019). This impact is not surprising, because PARylation is suggested to nucleate membranelles organelles, including stress granules (Isabelle et al, 2012; Alemasova & Lavrik, 2019; Ke et al, 2019). HuR has been shown to be located in membranelles organelles and is well known to form oligomers, which might be potentially regulated by PARylation, similar to hnRNPA1 (Isabelle et al, 2012). Therefore, it would not be surprising that an interplay exists between the covalent and non-covalent PARylation of HuR during apoptosis and potentially other systems. Our results shown in Fig 3D suggest that an interplay does indeed occur in HeLa cells because mutation of the HuR D226 residue or the HuR-PBS did not affect the association of HuR with PAR. The interplay between the non-covalent and the covalent PARylation of HuR, thus, may explain its differential role in modulating the survival or death of cells under normal or stress-induced conditions.

Our work, thus, has furthered our understanding of the role of HuR in apoptosis, showing that it is regulated by PARylation. Moreover, understanding the regulatory mechanism underlying the pivotal role of HuR in cell fate will bring a new hope to find therapies to overcome many diseases, such as numerous cancers, that are associated with the increased cytoplasmic localization of HuR.

# Materials and Methods

### Cell culture, transfection, and treatment

HeLa CCL-2 cells (American Type Culture Collection) were grown in a 5% $CO_2$ environment at 37°C in DMEM (Invitrogen) supplemented with 10% FBS (Invitrogen) and 1% penicillin/streptomycin antibiotics following the manufacturer's instruction (Invitrogen). The plasmid and siRNA were transfected as described by the Polyplus jetPRIME transfection protocol using 0.25–0.5 μg/ml and 50 nM/ml of plasmid and siRNA, respectively. Plasmid transfection was done when the HeLa cells reached 80% confluent, whereas siRNA transfection was done on 60% confluent HeLa cells. siRNAs were purchased from Ambion: siPARP1 (ID: s1097) and siPAPR2 (ID: 111561). For the staurosporine (STS) treatment (Sigma-Aldrich), HeLa cells

were incubated with different concentrations (0.1, 0.25, 0.5, and 1 $\mu$M) for 1.5 or 3 h. For epigallocatechin gallate (EGCG) treatment (Sigma-Aldrich), HeLa cells incubated with 500 $\mu$M for 2 h. Treatments were done 24 h post-transfection. For PARP inhibitor experiments, cells were treated with talazoparib 1 $\mu$M from Selleckchem (BMN673) for 24 h.

## Plasmid construction and protein purification

The GFP-HuR^WT and GST-HuR plasmids were generated as described previously with the GFP and GST added to the N-terminal of HuR (Mazroui et al, 2008). The GFP-HuR^PBmt and GST-HuR^PBmt plasmids (generated by mutating the histidine and arginine amino acids to alanines) and the point mutants GFP-HuR^D226A and GST-HuR^D226A plasmids (generated by mutating the Aspartic acid amino acid residue at 226^th position to alanine) were constructed by NorClone Biotech Laboratories. The GST, GST-HuR^WT, GST-HuR^PBmt, and GST-HuR^D226A recombinant proteins were generated by transforming BL21 with the respective plasmids. The expression of the proteins was induced by IPTG (0.5 mM for 4 h at 37°C) in a 1-liter culture. The bacteria were collected and lysed. The GST proteins were pulled down using glutathione Sepharose beads and processed as previously described (Brennan et al, 2000).

## Protein extraction and immunoblotting

Total cell extracts from the treated or untreated HeLa cells were prepared as described previously (von Roretz & Gallouzi, 2010). Briefly, cell extracts were lysed with mammalian lysis buffer (50 mM Hepes, pH 7.0, 150 mM NaCl, 10% glycerol, 1% Triton, 10 mM pyrophosphate sodium, 100 mM NaF, 1 mM EGTA, 1.5 mM MgCl2, 1 X protease inhibitor [Roche], and 0.1 M orthovanadate), and then, lysates were collected after centrifugation for 15 min at 12,879$g$ at 4°C. The extracts were then run on SDS–PAGE and transferred to nitrocellulose membranes (Bio-Rad) as described in Mazroui et al (2008) using the following antibodies: HuR (3A2 [Gallouzi et al, 2000], 1:1,000), cleaved caspase-3 (1:1,000; Cell Signaling), full-length PARP (1:1,000; Cell Signaling), GFP (JL-8, 1:1,000; Clonetech), and $\alpha$-tubulin (1:1,000; Abcam) as a loading control. Quantification of bands on Western blots was done using ImageJ (Fiji) software and normalized to $\alpha$-tubulin. Statistical analysis for significance was performed using GraphPad Prism 10 software with a two-tailed $t$ test.

## Subcellular fractionation and immunoblotting

The nuclear/cytoplasmic subcellular fractionation experiments were performed as described previously (van der Giessen et al, 2003). Briefly, cells were collected and washed once with ice-cold PBS. Subsequently, the cells were resuspended in 500 $\mu$l EBKL buffer (25 mM Hepes, pH 7.6, 5 mM MgCl2, 5 mM KCl, and 0.5% NP-40) and incubated for 15 min on ice. The cells were further lysed on ice by 30 strokes in a Dounce-type homogenizer using the tight pestle. The homogenate was subjected to a series of low- to high-speed centrifugations to separate the soluble cytoplasmic fraction (supernatant) from the nuclear fraction (pellet). The nuclear pellet was washed three times with EBMK buffer (no NP-40) by

centrifugation at 559$g$ for 3 min and resuspended in water containing 0.5% NP-40. Laemmli sample buffer was added to the samples and used for Western blot experiments using the following antibodies: GFP (JL-8, 1:1,000), HuR (3A2 [Gallouzi et al, 2000], 1:1,000), beta-actin (1:1,000; Invitrogen), H3 (1:1,000; Abcam), and $\alpha$-tubulin (1:1,000; Abcam). Quantification of bands on Western blots was done using ImageJ (Fiji) software and normalized to beta-actin. Statistical analysis for significance was performed using GraphPad Prism 10 software with a two-tailed $t$ test.

## Binding (dot/slot-blot) assay and peptide mapping experiments

These experiments were performed as described in Pleschke et al (2000). Briefly, GST, GST-HuR^wt, GST-HuR^pbmt, histone (positive control), and BSA (negative control), or peptides spanning the HuR protein (fragmented into 63 peptides; each is 20 amino acids in length) were dot-blotted directly onto a nitrocellulose membrane. The blot was then rinsed three times with TBST (Tris-buffered saline with 0.1% Tween-20 detergent) and incubated with radioactive pADPr (32P-pADPr) generated by automodified PARP1, washed, and probed for retention of the pADPr. After incubation for 1 h at room temperature with gentle agitation, the membrane was washed, dried, and subjected to autoradiography. Peptides/full-length proteins were incubated with SYPRO Ruby stain to demonstrate their integrity and event distribution.

## Immunoprecipitation

Immunoprecipitation experiments were performed as previously described (van der Giessen & Gallouzi, 2007; Cammas et al, 2014). Briefly, antibodies against anti-PAR 10H clone (Tulips), anti-TRN2 (van der Giessen & Gallouzi, 2007), and anti-PP32/PHAPI (Santa Cruz) were incubated with 60 $\mu$l of protein A Sepharose slurry beads (GE Healthcare) (washed and equilibrated in cell lysis buffer) for 4 h at 4°C. IP experiments for the drug-treated HeLa cells were done after treatment with STS or EGCG, respectively. For the plasmid-transfected cells, IP experiments were started 24 h after transfection and were done with Sera-Mag Protein A/G magnetic beads (Cytiva). First, 25 $\mu$l of magnetic beads was washed twice with the washing buffer (25 mM Tris, 0.65 M NaCl, and 0.05% Tween-20). Then, antibodies against anti-PAR 10H clone (Tulips) or mouse control IgG were incubated with the magnetic beads for 1 h at RT with gentle agitation. In both IPs, the remaining steps were the same. Beads were washed three times and incubated with 800 $\mu$g of total cell extracts from drug-treated or plasmid-transfected cells for overnight at 4°C. Beads were then washed (25 mM Tris, 0.15 M NaCl, 0.05% Tween-20, and 100 mM NaF) three times and eluted in 100 $\mu$l of 2x Laemmli dye by incubating the beads at 70°C for 10 min. Subsequently, 20 $\mu$l of the eluted samples was used for analysis by Western blot.

## RNA immunoprecipitation

HeLa cells transfected with GFP, GFP-HuR^WT, and GFP-HuR^PBmt were lysed, and 1 $\mu$l of RNaseOUT Recombinant Ribonuclease Inhibitor (Thermo Fisher Scientific) was added to each sample. The IP procedure as described before with magnetic beads was performed with an

antibody against GFP (JL-8, Living Colors). Before elution, 10% of the sample was saved for analysis by Western blot and 90% was incubated with 350 μl of RLT Plus buffer for 5 min before RNA extraction with RNeasy Plus Mini Kit (QIAGEN) as described in Mazaré et al (2020). Samples were eluted in 16 μl of RNA nuclease-free water, and all the eluted RNA was reverse-transcribed to cDNA using 5X iScript reagent (Bio-Rad). cDNA was diluted 10-fold and used to detect mRNA levels of caspase-9 and ProTα using Power SYBR Green PCR Master Mix (Thermo Fisher Scientific). A relative gene expression change was analyzed and normalized to GFP-transfected cells by RT–qPCR. Primers used for qRT-PCR are as follows: caspase-9 (F: 5′-GTTTGAGGACCTTCGACCAG-3′, R:5′-GCATTAGCGACCCTAAGCAG-3′), ProTα (F: 5′-CTGCTAACGGGAATGCTGA-3′, R:5′-TCGACATCGTCATCCTCATC-3′).

### Immunofluorescence staining of cells

Immunofluorescence was performed as previously described (von Roretz et al, 2011). IF experiments for the drug-treated HeLa cells were performed after 1.5 or 1 h of incubation with STS or EGCG, respectively. For the plasmid-transfected cells, IF experiments were performed 24 h after transfection. Briefly, HeLa cells cultured on 35-mm imaging dishes with glass coverslip bottom (ibidi) were rinsed twice in PBS, fixed with 4% PFA (Sigma-Aldrich) in PBS for 10 min, and then permeabilized with 0.1% Triton X-100 in PBS/goat serum for 1 h at room temperature. After permeabilization, cells were incubated with primary antibodies against HuR/3A2 (1:1,000) in 1% normal goat serum/PBS at 4°C overnight. The cells were then incubated with the secondary antibody (Alexa Fluor 488) and 4′,6-diamidino-2-phenylindole (DAPI) (for nuclear staining). IF plates were observed at room temperature with a 63X oil objective Zeiss Axiovision 3.1 microscope, and an AxioCam HR (Zeiss) digital camera was used for immunofluorescence photography.

### Annexin V–Cy5/PI assay

HeLa cells after 24 h post-transfection for siRNA or plasmids were collected by trypsinizing the plate. Cell pellets were then processed as described by the apoptosis detection reagent kit protocol (ab14147 & ab14084; Abcam). Apoptotic and necrotic cells were identified by annexin V–Cy5 and propidium iodide (PI) staining, respectively, using the flow cytometry analyzer (FACSCanto II). The flow cytometry work was performed in the Flow Cytometry Core Facility for flow cytometry and single-cell analysis of the Life Science Complex. The data were then analyzed using FlowJo software.

### Quantitative RT–qPCR

RNA was extracted from cell extracts using TRIzol reagent (Invitrogen) according to the manufacturer's instructions. One microgram of total RNA was reverse-transcribed using the 5X iScript reagent (Bio-Rad) according to the manufacturer's protocol. Each cDNA sample was diluted 20-fold and used to detect the mRNA levels of PARP1, PARP2, and GAPDH (used as a loading control) using the SsoFast EvaGreen reagent (Bio-Rad Laboratories). The relative expression level was calculated using the $2^{-\Delta\Delta Ct}$ method, where ΔΔCt is the difference in Ct values between the target and reference genes (GAPDH). Primers used for qRT-PCR are as follows: PARP1 (F: 5′-CCCAGGGTCTTCGAATAG-3′,

R: 5′-AGCGTGCTTCAGTTCATAC-3′), PARP2 (F: 5′-GGAAGGCGAGTGCTAAAT-GAA-3′, R: 5′-AAGGTCTTCACAGAGTCTCGATTG-3′), GAPDH (F:5′-AAGGT-CATCCCAGAGCTGAA-3′, R: 5′-AGGAGACAACCTGGTCCTCA-3′).

## Data Availability

Any additional information required to reanalyze the data reported in this study is available from the corresponding contact upon request.

## Supplementary Information

## Acknowledgements

This work was funded by a CIHR operating grant (MOP-142399) and a CIHR project grant (PJT-159618) to I-E Gallouzi. This work was also supported by a BAS/1/1035-01-01 baseline and KAUST Smart-Health Initiative (KSHI) funding to I-E Gallouzi. K Ashour was funded by a scholarship received from the Faculty of Applied Medical Sciences of Taibah University/the Ministry of Higher Education. S Mubaid was funded by three scholarships received from the Faculty of Medicine of McGill University.

### Author Contributions

K Ashour: conceptualization, data curation, formal analysis, validation, investigation, methodology, and writing—original draft.
S Sali: formal analysis, validation, investigation, visualization, methodology, and writing—review and editing.
AH Aldoukhi: conceptualization, validation, visualization, and methodology.
D Hall: conceptualization, formal analysis, validation, visualization, and methodology.
S Mubaid: data curation, validation, visualization, and methodology.
S Busque: data curation, validation, visualization, and methodology.
XJ Lian: data curation, validation, visualization, and methodology.
J-P Gagné: conceptualization, formal analysis, validation, visualization, and methodology.
S Khattak: conceptualization, formal analysis, validation, visualization, and writing—review and editing.
S Di Marco: conceptualization, formal analysis, validation, investigation, visualization, and writing—review and editing.
GG Poirier: conceptualization and supervision.
I-E Gallouzi: conceptualization, formal analysis, supervision, funding acquisition, investigation, visualization, and writing—original draft, review, and editing.

### Conflict of Interest Statement

The authors declare that they have no conflict of interest.

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
