## [Reviewer comments · Life Science Alliance]

Life Science Alliance

pADP-ribosylation Regulates the Cytoplasmic Localization, Cleavage and Pro-apoptotic Function of HuR

Kholoud Ashour, Sujitha Sali, Ali Aldoukhi, Derek Hall, Souad Mubaid, Sandrine Bousque, Xian Lian, Jean-Phillipe Gagné, Shahryar Khattak, Sergio Di Marco, Guy Poirier, and Imed Gallouzi

DOI: <https://doi.org/10.26508/lsa.202302316>

Corresponding author(s): Imed Gallouzi, King Abdullah University of Science and Technology

Review Timeline:

Submission Date:	2023-08-13
Editorial Decision:	2023-09-14
Revision Received:	2024-02-03
Editorial Decision:	2024-03-13
Revision Received:	2024-03-19
Accepted:	2024-03-20

Transaction Report:

September 14, 2023

Re: Life Science Alliance manuscript #LSA-2023-02316-T

Prof. Imed Eddine Gallouzi
King Abdullah University for Sciences and Technology
BESE & KSHI
Saudi Arabia

Dear Dr. Gallouzi,

Thank you for submitting your manuscript entitled "pADP-ribosylation Regulates the Cytoplasmic Localization, Cleavage and Pro-apoptotic Function of HuR" to Life Science Alliance. The manuscript was assessed by expert reviewers, whose comments are appended to this letter. We invite you to submit a revised manuscript addressing the Reviewer comments.

Thank you for this interesting contribution to Life Science Alliance. We are looking forward to receiving your revised manuscript.

Sincerely,

B. MANUSCRIPT ORGANIZATION AND FORMATTING:

Reviewer #1 (Comments to the Authors (Required)):

In this study, the authors identify the non-covalent association of RNA-binding protein HuR to pADP-ribose (PAR) and report that HuR-PAR promotes its nuclear localization and integrity. Destruction of PARP1, a major enzyme in the synthesis of PAR, during apoptosis reduces HuR-PAR, and the free HuR cannot be efficiently returned to the nucleus, accumulates in the cytoplasm, and is cleaved by caspases.

Although covalent PARylation of HuR was previously reported, non-covalent interaction of HuR with PAR is novel and interesting. Some deeper analysis is suggested in order to strengthen the authors' model.

Main comments

1. At the start of Results, it is unclear what prompted the authors to examine non-covalent association of PAR to HuR. It would be helpful if the authors offered a more explicit rationale.
2. More biochemical detail of HuR PARylation in this paradigm would be helpful. What ratio of HuR is PARylated non-covalently? how about covalently? The PAR residues associated non-covalently with HuR, are they single PAR molecules or concatemers of PAR?
3. The data in Figure 4 suggest that in the absence of stress, the PAR-HuR association contributes to retaining HuR in the nucleus. To understand this regulation in context, the authors should study the subcellular localization of HuRwt and HuRPbmt in response to a variety of stresses (not only STS). It would be particularly informative to study the subcellular localization of each GFP-HuR fusion protein (wt and Pbmt) over a range of stresses, including mild, moderate, and apoptosis-inducing stress levels.
4. To solidify the authors' model, it will be important to test if the various stresses (as requested above) trigger a loss of PAR association with the endogenous HuR protein.
5. Does HuR-PAR in this model bind HuR target mRNAs differently from HuR alone?
6. In Figure 4, was GFP added N-terminal to HuR? Presumably this is the case, as the authors detect GFP-CP1. This detail should be specified in the Results.
7. The author's proposed notion that HuR might bind PAR covalently in some instances (e.g., after treatment with LPS) and it would bind non-covalently in other instances (e.g., after treatment with apoptotic STS) is very interesting. In this regard, does the covalent PARylation of HuR (at D226 presumably) affect the non-covalent PARylation of HuR (at the PBS)? And vice versa? The authors have the tools and expertise to examine these questions. The result of analyzing the interplay between covalent and non-covalent PARylation would likely transcend HuR and be relevant to other proteins more broadly.
8. In the model, for clarity, the authors are advised to modify the right panel (Apoptotic condition) and remove the HuR-TRN2 complexes, since according to the authors' hypothesis, TRN2 does not appear to bind HuR when HuR is not associated with PAR.
9. In addition, when describing the model, the authors are advised to acknowledge alternative explanations. Given that so many cellular proteins are PAR-modified and PAR-associated, loss or inhibition of PARP could affect the function of other proteins in the complex machineries that either retain HuR in nucleus or cytosol, or modulate HuR levels. In other words, the import of the HuR-PAR complex by TRN2 may be only one of the ways how PARP1 controls cytosolic HuR levels.

Reviewer #2 (Comments to the Authors (Required)):

In this manuscript the authors extend their previous work on the relationship between HuR, cleavage, PARPs/parylation and apoptosis. The experimental design is straightforward, well-controlled and appropriate for the questions being addressed. The conclusions are generally well-supported by the data and I believe that the work will be of interest to the field. I only have a few

suggestions to polish the presentation and clarify a few points:

Points:

1. Pg. 7: It is unclear what the authors mean by 'we observed, similarly, a trend in HuR cleavage in cells depleted of PARP2'. Inconsistent with that conclusion, Fig. 2A, lane 3 shows very little, if any, cleavage of HuR compared to the PARP1 KD in lane 2 - In fact, its very similar to control conditions. The associated graph on the right does not appear to reflect the gel data shown on the left and does not have statistical significance. Please clarify. Also was the cleavage truly 'further significantly increased in the double knockdown (concluded on page 7)' when compared to the KD of PARP1 alone? Finally, the caspase 3 cleavage on the graph at the right should include statistical analysis to support the claim of a significant increase in cleavage in the double knockdown.
2. Fig. 2B would also benefit from statistical analysis.
3. Fig. 4B: the data appears to indicate that the HuR Pbm1 isoform reduces its association with TRN2 rather than 'loses' its association with TRN2 as concluded by the authors. Please clarify.

Very minor points:

1. Throughout the manuscript, some references are spaced improperly (for example 'apoptosis (Pleschke et al, 2000; Wei...) is spaced properly where others are lacking a space between the parenthesis and the previous word (e.g. 'apoptosis(Pleschke...)' Since the manuscript may not be type edited prior to publication, the authors may wish to go through an edit this aspect of the manuscript.
2. Pg. 6 change 3h hours to 3 hours.
3. Optimally, please add stats to Fig. 5A/B (graphs on the right).

Reviewer #3 (Comments to the Authors (Required)):

This is a strong report from a very strong group that studies both the RNA binding protein HuR and its subcellular localization in relation to its various functions under moderate and severe stress. The manuscript is important to the field and should be respectfully published. A couple of items should be addressed however:

- 1) The exact link of subcellular localization, cleavage form of HuR and stress in relation to function is not directly shown in an experiment. Some experiments are descriptive and suggestive of this connection, but could the investigators provide any more direct evidence putting their fig. 6 schematic together.
- 2) IF is strong data, but do the investigators have any subcellular localization western blots with fractions to provide support for the IF data.
- 3) In some instances, showing the entire blot would help the field in seeing the different cleavage sizes/products.

Point-by-point rebuttal to reviewers' comments

We thank the reviewers for their detailed review of our manuscript and for their comments and suggestions. Having addressed these comments/suggestions, we believe that the revised manuscript is significantly improved, increasing the impact of the article.

Reviewers' comments provided below are shown in ***bold, italics font*** while our responses are in normal font.

Reviewer #1

(Comments for the Author):

Main comments

1) At the start of Results, it is unclear what prompted the authors to examine non-covalent association of PAR to HuR. It would be helpful if the authors offered a more explicit rationale.

We thank the reviewer for his comment. We have amended our text at the start of the Results section to explain the rationale more clearly for examining the non-covalent association of PAR to HuR. As described on page 6 of the revised manuscript we have explained that the rationale for doing so is based on the fact that the activity/function of numerous RNA-binding proteins as well as the activation of several intracellular pathways including cell death is mediated by the non-covalent binding of proteins to PAR (PMID: 23268355, 32029452, 26673700).

2) More biochemical detail of HuR PARylation in this paradigm would be helpful. What ratio of HuR is PARylated noncovalently? how about covalently? The PAR residues associated non-covalently with HuR, are they single PAR molecules or concatemers of PAR.

We thank the reviewer for this comment. To the best of our knowledge, the determination of the ratio of covalent versus non-covalent PARylation of proteins has never been reported. We believe that attempting to estimate the ratio of covalent versus non-covalent PARylation of HuR would be extremely difficult to accomplish since both scenarios are likely co-occurring in cells. In our manuscript, we identified a PAR binding motif (PBM) within HuR which mediates its localization and function during apoptosis. We are unsure, at this time, if the PAR residues associated non-covalently to HuR are single ADP-ribosyl moieties or PAR polymers. Although the exact mechanism through which PBMs bind non-covalently to PAR is not well understood, it has been hypothesized that this motif interacts with PAR by binding in between the second phosphate of one ADP-ribosyl moiety and the first phosphate of the next (PMID: 34302489, 31842403). This thus would suggest that HuR, containing a PBM, most likely

does not bind to single PAR molecule. This conclusion is supported by our *in vitro* data demonstrating that mutations of this motif prevented the binding of HuR to ADPr polymers generated by auto-modified PARP1 (Fig. 3C). A statement to this effect has been included in the discussion of the revised manuscript (page 14).

3) The data in Figure 4 suggest that in the absence of stress, the PAR-HuR association contributes to retaining HuR in the nucleus. To understand this regulation in context, the authors should study the subcellular localization of HuR^{wt} and HuR^{Pbmt} in response to a variety of stresses (not only STS). It would be particularly informative to study the subcellular localization of each GFP-HuR fusion protein (wt and Pbmt) over a range of stresses, including mild, moderate, and apoptosis-inducing stress levels.

We thank the reviewer for this comment. In order to identify conditions reflecting mild, moderate and apoptosis-inducing stress levels we treated HeLa cells in a dose-dependent manner with different concentrations of STS (0.1 μ M, 0.25 μ M, 0.5 μ M & 1 μ M). We observed, in doing so, that the cleavage of PARP1, HuR and Caspase-3 increased in a dose-dependent manner. In performing these experiments, we observed that treatment with 0.1 μ M and 0.25 μ M STS represent mild apoptotic conditions since they only slightly increased, albeit non-significantly, the cleavage of HuR. Additionally, although the cleavage of HuR, PARP1 and Caspase-3 is statistically significant starting at a 0.5 μ M concentration (representing moderate conditions), the levels observed were much less than those obtained with 0.1 μ M (representing severe stress) (Fig. 1A of the revised manuscript). Having established these conditions we subsequently showed that the cytoplasmic localization of GFP-HuR is correlated with the severity of the apoptosis-inducing stress level. Although GFP-HuR^{wt} is nuclear under mild conditions (0.1 μ M and 0.25 μ M concentration), it begins to become apparent in the cytoplasm of cells under moderate conditions (0.5 μ M concentration), reaching maximal levels under severe apoptotic conditions (0.1 μ M) (Fig. 4B, Supp. Fig.6 of the revised manuscript). The effect of these apoptotic conditions on the localization of GFP-HuR^{wt} is identical to what we observed for endogenous HuR in cells treated with STS (Fig 1B and Fig S2 of the revised manuscript). Unlike GFP-HuR^{wt}, the cytoplasmic accumulation of GFP-HuR^{PBmt}, which is evident in untreated cells, is not further affected by treating cells over a range of apoptotic-inducing stress conditions. These results, thus, demonstrate that the binding of HuR to PAR, via its PBM, is required for retaining HuR to the nucleus of cells.

In addition to the various STS-inducing stress conditions, we also investigated the localization of GFP-HuR^{wt} and GFP-HuR^{PBmt} in cells treated with Epigallocatechin-3-gallate (EGCG). EGCG has been previously shown to induce apoptosis in a variety of cancer cells (PMID: 28884125, 29588626) by inhibiting, in part, the activity of PARP16 (PMID: 28698806). We observed that while treatment with EGCG activated apoptosis, it did so without inducing cleavage of HuR nor affecting its cytoplasmic accumulation (Fig 1A and B). The cytoplasmic accumulation of GFP-HuR^{PBmt}, which is evident in untreated cells, is not further affected by treatment with EGCG.

Collectively, therefore, our data suggests that the nuclear retention of HuR, which is observed under non-apoptotic conditions, is mediated by its ability to non-covalently interact with PAR.

4) To solidify the authors' model, it will be important to test if the various stresses (as requested above) trigger a loss of PAR association with the endogenous HuR protein.

We thank the reviewer for this comment. We have performed these experiments and have shown that while HuR associates to PAR under mild apoptotic and untreated conditions (where HuR is completely localized to the nucleus of cells and is not cleaved), it loses its binding to PAR under moderate and severe apoptotic conditions where PARP1 and HuR are cleaved and HuR, further, is localized to the cytoplasmic of these treated cells (Fig 1A-C of the revised manuscript). Interestingly, we also demonstrate that the association of HuR to PAR is not affected by treatment of cells with EGCG, correlating with its localization in these cells, despite the activation of apoptosis (as evidenced by the cleavage of caspase-3) (Fig 1A-C of the revised manuscript).

5) Does HuR-PAR in this model bind HuR target mRNAs differently from HuR alone?

We thank the reviewer for this comment. We have provided new data in the revised manuscript demonstrating that although the non-covalent binding of HuR to PAR affects its cellular localization it does not affect its ability to associate to mRNA targets including the previously identified *caspase-9* and *prothymosin a* mRNA (Supp. Fig. 4 of the revised manuscript).

6) In Figure 4, was GFP added N-terminal to HuR? Presumably this is the case, as the authors detect GFP-CP1. This detail should be specified in the Results.

We thank the reviewer for this comment. The GFP was indeed added N-terminal to HuR. A statement to this effect was included on page 9 of the Results section of the revised manuscript as well as page 17 of the Material and Methods section.

7) The author's proposed notion that HuR might bind PAR covalently in some instances (e.g., after treatment with LPS) and it would bind non-covalently in other instances (e.g., after treatment with apoptotic STS) is very interesting. In this regard, does the covalent PARylation of HuR (at D226 presumably) affect the non-covalent PARylation of HuR (at the PBS)? And vice versa? The authors have the tools and expertise to examine these questions. The result of analyzing the interplay between covalent and non-covalent PARylation would likely transcend HuR and be relevant to other proteins more broadly.

We thank the reviewer for this comment and for suggesting this experiment that helped us clarify our conclusions. To address this comment, we assessed if the mutation of the D226 residue affects the non-covalent PARylation of HuR and, vice versa if the mutation of the PAR binding motif of HuR affects its covalent PARylation. We thus immunoprecipitated PAR from HeLa cells transfected with GFP, GFP-HuR^{WT}, GFP-HuR^{PBmt} (containing a mutation in the PAR binding motif) and GFP-HuR^{D226A} (containing a mutation at the D226 residue) followed by western blot analysis with anti-GFP (Fig. 3D of the revised manuscript). Our new experiments (N=3) clearly show that GFP-HuR^{WT} and GFP-HuR^{PBmt} were similarly associated to PAR suggesting that the mutation of the PAR binding motif did not hinder the covalent PARylation of HuR. Mutation of the D226 residue, on the other hand, increased the non-covalent association of PAR to HuR. This effect is likely because the GFP-HuR^{D226A} mutant, while not cleaved, prevents caspase-mediated apoptosis (Fig. 3D of the revised manuscript). Together, these results reveal that an interplay between the non-covalent and covalent PARylation of HuR may explain its differential role in modulating the survival or death of cells under normal or stress-induced conditions. The text of the result and discussion section were amended to reflect these new data.

8) In the model, for clarity, the authors are advised to modify the right panel (Apoptotic condition) and remove the HuR-TRN2 complexes, since according to the authors' hypothesis, TRN2 does not appear to bind HuR when HuR is not associated with PAR.

We thank the reviewer for this comment. We have, as suggested by the reviewer, modified the model presented in Fig 6 of the revised manuscript by removing the HuR-TRN2 complexes in the right panel (Apoptotic conditions).

9) In addition, when describing the model, the authors are advised to acknowledge alternative explanations. Given that so many cellular proteins are PAR-modified and PAR-associated, loss or inhibition of PARP could affect the function of other proteins in the complex machineries that either retain HuR in nucleus or cytosol, or modulate HuR levels. In other words, the import of the HuR-PAR complex by TRN2 may be only one of the ways how PARP1 controls cytosolic HuR levels.

We thank the reviewer for this comment. We have amended the text on page 14 of the revised manuscript to include a statement indicating that the import of the HuR-PAR complex by TRN2 may be only one of the ways how PARP1 controls cytosolic HuR levels.

Reviewer #2

1) Pg. 7: It is unclear what the authors mean by 'we observed, similarly, a trend in HuR cleavage in cells depleted of PARP2'. Inconsistent with that conclusion, Fig. 2A, lane 3 shows very little, if any, cleavage of HuR compared to the PARP1 KD in lane 2 - In fact, its very similar to control conditions. The associated graph on the

right does not appear to reflect the gel data shown on the left and does not have statistical significance. Please clarify. Also was the cleavage truly 'further significantly increased in the double knockdown (concluded on page 7)' when compared to the KD of PARP1 alone? Finally, the caspase 3 cleavage on the graph at the right should include statistical analysis to support the claim of a significant increase in cleavage in the double knockdown.

We thank the reviewer for their comment. We have repeated these experiments in order to gain more clarity on the effect of knocking down PARP1 and PARP2 on the cleavage of HuR and the induction of apoptosis. Our new data, shown in Fig 2 of the revised manuscript clearly show that although the knockdown of PARP1 significantly induced the cleavage of HuR (reflecting its localization to the cytoplasm of cells) and the induction of apoptosis, these effects were not observed in cells depleted of PARP2. The cleavage of HuR, furthermore, was further significantly increased in the double knockdown when compared to the knockdown of PARP1 alone. We have lastly included statistical analyses to support the notion that the cleavage of caspase 3 is significantly increased in the double knockdown conditions.

2. Fig. 2B would also benefit from statistical analysis.

We thank the reviewer for this comment. As requested, we have included statistical analyses in Fig 2D of the revised manuscript (Fig 2B of the original manuscript).

3. Fig. 4B: the data appears to indicate that the HuR Pbmt isoform reduces its association with TRN2 rather than 'loses' its association with TRN2 as concluded by the authors. Please clarify.

We thank the reviewer for this comment. We have modified our text (page 10 of the revised manuscript) to indicate that the association is reduced rather than lost.

Very minor points:

1) Throughout the manuscript, some references are spaced improperly (for example 'apoptosis (Pleschke et al, 2000; Wei...) is spaced properly where others are lacking a space between the parenthesis and the previous word (e.g. 'apoptosis(Pleschke...)' Since the manuscript may not be type edited prior to publication, the authors may wish to go through an edit this aspect of the manuscript.

We thank the reviewer for pointing this out to us. We have corrected the improper spacing accordingly.

2) Pg. 6 change 3h hours to 3 hours.

We thank the reviewer for pointing this out to us. We have amended our text accordingly.

3) Optimally, please add stats to Fig. 5A/B (graphs on the right).

We thank the reviewer for this comment. As requested, we have included statistical analyses in Fig 5A and B of the revised manuscript.

Reviewer #3

1) The exact link of subcellular localization, cleavage form of HuR and stress in relation to function is not directly shown in an experiment. Some experiments are descriptive and suggestive of this connection, but could the investigators provide any more direct evidence putting their fig. 6 schematic together.

We thank the reviewer for this comment. As mentioned above we show, in the revised manuscript, that overexpressing GFP-HuR^{PBmt}, containing a mutation in the PAR-binding motif, increased the cytoplasmic accumulation of HuR as well as its cleavage and the induction of apoptosis. We further show that mutation of the D226 residue, which prevents its cleavage and does not induce apoptosis, surprisingly increased the association of HuR to PAR. These results thus link HuR interaction with PAR (or a lack of) as a key determinant that decided the fate of apoptotic cells (Fig 3C and D, Fig 4A and B, Fig 5 of the revised manuscript.).

2) IF is strong data, but do the investigators have any subcellular localization western blots with fractions to provide support for the IF data.

As suggested by the reviewer, we have included subcellular localization data to support the IF data contained in the original manuscript (Fig S3B and C as well as Fig S5 of the revised manuscript).

3) In some instances, showing the entire blot would help the field in seeing the different cleavage sizes/products

We thank the reviewer for pointing this out to us. We now show, in the revised manuscript, the entire western blots in order to see the different cleavage sizes/products.

March 13, 2024

RE: Life Science Alliance Manuscript #LSA-2023-02316-TR

Dr. Imed Eddine Gallouzi
King Abdullah University of Science and Technology
BESE & KSHI
Al-Jazri Building, bld 4, level 3 (spine side), office 3337
Thuwal 23955-6900
Saudi Arabia

Dear Dr. Gallouzi,

Thank you for submitting your revised manuscript entitled "pADP-ribosylation Regulates the Cytoplasmic Localization, Cleavage and Pro-apoptotic Function of HuR". We would be happy to publish your paper in Life Science Alliance pending final revisions necessary to meet our formatting guidelines.

- please be sure that the authorship listing and order is correct
- please add ORCID ID for the corresponding author -- you should have received instructions on how to do so
- please add an Author Contributions section to your main manuscript text
- Figures S2, S5, and S6 have no panels, so please revise their legends
- Figure 6 can instead be uploaded as a Graphical Abstract, rather than as a figure. This is up to you.

Figure Checks:

- Figure 4B and S6 contain repeat images. Please indicate clearly in the Figure S6 legend that it is showing the unmerged images from the same experiments shown in Figure 4B.
- please provide the original blots used to make the left panel of Figure 4C as Source Data
- please add scale bars to Figures S1B and S2

A. FINAL FILES:

B. MANUSCRIPT ORGANIZATION AND FORMATTING:

Sincerely,

Reviewer #1 (Comments to the Authors (Required)):

I appreciate the authors' thorough responses. My concerns have been addressed in full.

Reviewer #2 (Comments to the Authors (Required)):

The authors have effectively addressed my comments on the previous version of the manuscript. I find the revised manuscript to be improved and convincing.

Reviewer #3 (Comments to the Authors (Required)):

The authors do an adequate job of addressing the reviewers' comments.

March 20, 2024

RE: Life Science Alliance Manuscript #LSA-2023-02316-TRR

Dr. Imed Eddine Gallouzi
King Abdullah University of Science and Technology
BESE & KSHI
Al-Jazri Building, bld 4, level 3 (spine side), office 3337
Thuwal 23955-6900
Saudi Arabia

Dear Dr. Gallouzi,

Thank you for submitting your Research Article entitled "pADP-ribosylation Regulates the Cytoplasmic Localization, Cleavage and Pro-apoptotic Function of HuR". It is a pleasure to let you know that your manuscript is now accepted for publication in Life Science Alliance. Congratulations on this interesting work.

DISTRIBUTION OF MATERIALS:

Again, congratulations on a very nice paper. I hope you found the review process to be constructive and are pleased with how the manuscript was handled editorially. We look forward to future exciting submissions from your lab.

Sincerely,
